# Private Long-Term Care Insurance Decision: The Role of Income, Risk Propensity, Personality, and Life Experience

**DOI:** 10.3390/healthcare9010102

**Published:** 2021-01-19

**Authors:** Shu-Chuan Jennifer Yeh, Wen Chun Wang, Hsueh-Chih Chou, Shih-Hua Sarah Chen

**Affiliations:** 1Department of Business Management, National Sun Yat-sen University, Kaohsiung City 804, Taiwan; wanchinwang122@gmail.com (W.C.W.); hcchou@vghks.gov.tw (H.-C.C.); 2Kaohsiung Veterans General Hospital, Kaohsiung City 813, Taiwan; schen9215@gmail.com; 3Division of Social Sciences, The University of Chicago, Chicago, IL 60637, USA

**Keywords:** long-term care insurance, consumer decision making, risk propensity, personality, discretionary income

## Abstract

The rising aging population contributes to increased caregiver burden and a greater need for long-term care services, thereby posing stronger financial burden. The current study aimed to examine the effect of income, risk-taking propensity, personality traits, and life experience on the ownership of and intention to own private long-term care insurance (LTCI). Primary data were collected from 1373 registered nurses with a minimum of two years of full-time working experience. Multinomial logistic regression was used to examine the relationships between ownership of LTCI and personal discretionary income, risk propensity, openness to experience, and life experience. Personal discretionary income was a crucial positive indicator in predicting ownership of LTCI. Higher risk-taking propensity was found to be negatively related to both currently own and future intention to own private LTCI. Participants who currently live with elders and who agree to caregiving responsibilities with government-provided cash allowance showed future intention to purchase LTCI. Little evidence was found for an association between life experience and future intention to own LTCI. Income, risk-taking propensity, and personality traits differ in their impact on ownership of and future intention to own LTCI. Our results provide policy makers with a better understanding of the forces driving demand in the private LTCI market, as well as the accompanying implications for public LTCI.

## 1. Introduction

The aging problem has urged countries worldwide to take initiative in establishing long-term care systems. Developed countries such as Germany, Japan, the United States, and the United Kingdom have acted early on to establish public and private long-term care insurance (LTCI) programs, in which both the government and private sector have played a fundamental role in insuring for LTC expenditures [1,2,3]. Despite the difference in operating mechanisms, public and private plans share the common mission to reduce uncertainty and risks associated with long-term care expenditures for the elders and disabled of our society.

The role of caregivers is key in LTC services. Oftentimes, this part is taken up by family members. In countries such as Germany, France, and Austria, adult children are legally compelled to assist ailing parents who have exhausted their own financial resources [4]. Even in the absence of legal obligation, strong moral responsibility for adult children to assist parents in need is found in many cultures, as is the case of Japan and Taiwan. According to the Taiwan Association of Family Caregivers 2019 report [5], 65% of caregivers were family members. Yet, as our society shifts toward lower birthrates, nulear families, a rising population of unmarried adults, and delayed marriage or childbearing age, the number of potential family caregivers is decreasing, thereby underscoring the importance of repositioning our dependence on nonfamily caregivers to provide home care, day care, or institutional care.

The potential to provide care in either public or private sector is heavily reliant on strong financial support. In response to the augmenting pressure of financing public insurance, cost shifting toward private individuals will continue. Private LTC insurance therein becomes a potential funding source, easing the burden of LTC costs on individuals. Yet, given the high cost of private LTCI and the overwhelmingly diverse coverage from which elders select, relatively few of the elderly and near-elderly have chosen to purchase LTCI [6,7]. For example, about 8% of elderly Americans have some form of private insurance to cover their long-term care needs [8,9]. Germany reported similar percentages, while the UK indicated that private insurance accounted for a negligible share of total LTC expenditure in the UK [10]. In Taiwan, there were 567,000 effectively private LTCI policies in 2016, which accounted for 2.4% of the population [11].

Several recent studies have supported that certain individual characteristics or competencies are associated with higher rates of private LTCI purchasing. In particular, individuals with better cognitive capacity, jobs related to finance, or ability to manipulate numeric information (as defined by one’s quantitative reasoning skills and propensity to implement them during decision-making tasks [12]) have been found to be more likely to buy LTCI [13,14,15]. Another study found that respondents who are subjected to narrow framing were less likely to buy LTCI [6]. Most of the aforementioned studies examined single dimensions of the determinants of LTCI purchase or ownership. However, making a better choice for LTCI is likely dependent on several individual characteristics, such as life experience, personal traits, and economic status [16,17]. Therefore, by delineating the way individual characteristics are associated with the LTCI choice, this study intended to provide a more comprehensive portrait of individual behavior in such decision context.

## 2. Background and Literature Review

### 2.1. Long-Term Care in Taiwan

The increasing aging population and workload of working-aged adults facilitate the needs of long-term care. In 2019, the aging population in Taiwan was 14.94%, thereby reaching an aged society, and is expected to escalate to 25% by 2026 [18]. Taiwan’s population aged 65 and above is growing at a much faster rate than many other countries—an estimation of only eight years to advance from the “aged society” stage to the “super-aged society” stage, in comparison to 11 years for Japan, 14 years for the United States, 29 years for France, and 51 years for the United Kingdom [19]. The growing rate of Taiwan, however, has been predicted to be similar to that of Singapore (seven years) and South Korea (eight years). Taiwan, Hong Kong, and South Korea will have 4.5, 3.8, and 4.6 working-age adults per elderly person in 2020, but 2.8, 2.3, and 2.7, respectively, by 2030, and just 1.5 apiece by 2050 [19].

Long-term care is not only for the elderly but also for the disabled. The disabled population has continuously risen from 1.10 million in 2017 to 1.16 million in 2019. The average number of chronic diseases was 1.16 for aged 65 and above [20], with a majority of caregivers either family members (65%) or foreign caregivers (28%). Among family caregivers, 75% were women (daughters, daughters-in-law, and spouses) and approximately 20% were younger elders taking care of older elders [5]. Average care hours were 13.5 daily hours. Approximately 2.3 million jobs were influenced as a result of providing care to family members in need of long-term care services. In summary, the aging phenomenon, increasing disabled population, and caregivers’ burden facilitate the need of public long-term care services [21].

The Long-term Care Services (LTCS) was initiated in Taiwan in 2017, as a response to the emergent aging population. The need for LTCS was anticipated at about 510,000 persons, but only 33% of this estimated population utilized this service in 2017 (called LTC Act 1.0). In 2017, the LTCS (called LTCS Act 2.0) further expanded its coverage from eight items to 17 items, thereby increasing the estimated need to 738,000 persons [22].

The LTCS provides coverage based on individuals’ health condition instead of income level. The Centers for Medicare and Medicaid Services (CMS) in Taiwan assigns a specialist to evaluate the applicant’s health status based on several indexes such as Activities of Daily Living (ADL), Instrumental Activities of Daily Living (IADL), Frailty Index (from the Study of Osteoporotic Fraction (SOF)), etc., and classifies an applicant into one of eight levels. To qualify for LTCS, applicants must be receive a classification of level 2 or above and meet at least one of the following conditions: be of age 65 years or older (or age 55 years or older if a part of the indigenous population), hold a certificate of disability, or be of age 50 years or older with dementia. In general, the government-provided LTCS cannot fulfill the demand for needed care without imposing strong financial burden on the government. This gap between the needed and the granted services may be filled by private long-term care insurance or by out of pocket.

The financial budget for LTCS in Taiwan comes mainly from taxes. The funding increased by 3.2 times, from US$ 343 million in 2017 to US$ 1.1 billion in 2018. The total budget will be set at around US$ 2.42 billion by year 2026—less than 0.3% of Taiwan’s entire GDP (Gross Domestic Product) on long-term care [22], which is relatively small compared to the Netherlands (3.7%), Norway (3.3%), Sweden (3.2%), Japan (1.8%), Germany (1.5%), UK (1.4%), and US (0.5%) [23]. Therefore, finding other ways to pay for LTCS would increase the utilization of LTCS and fulfill the need for LTCS, as well as decrease the financial burden from government. Private LTCI is another way of risk sharing—by lowering the out-of-pocket price for long-term care at the time of purchase and by smoothing LTCS payments across individuals and time [24].

### 2.2. Personal Discretionary Income

Individuals consider both price and income when deciding whether to purchase a certain good [25]. According to consumer theory, health insurance is expected to be a normal good with a positive income elasticity of demand, thereby implying that the poor should be less likely to insure [26]. Higher income and assets strengthen one’s ability to purchase LTCI. As evidence has shown over the past 25 years, LTCI ownership has been dominated by the highest earners and the wealthy [27]; furthermore, income is representative of one’s ability to afford a LTCI plan [28]. Thus, individuals’ financial status is a crucial factor that influences the purchase of LTCI. 

**Hypothesis** **1.**
*Individuals with more discretionary income are more likely to purchase LTCI or demonstrate intention to purchase LTCI.*


### 2.3. Risk-Taking Propensity

According to Sitkin and Weingart (1995) [29], risk propensity is defined as an individual’s current tendency to take or avoid risks. Risk-taking propensity is the tendency to engage in behavior that bears the chance of losses (e.g., financial losses, physical harm) as well as gains (e.g., financial gains, excitement) [30]. When facing risks, individuals also differ in their ways of dealing with the risks [31]. Economists have classified people according to these differences into three categories: (1) the risk averse: individuals who prefer certainty and are willing to pay for certainty even when the cost of certainty exceeds the probable loss; (2) the risk preferers: individuals who are comfortable facing uncertainty and tend to accept a degree of uncertainty even if offered the opportunity to pay for certainty; (3) the risk neutral: individuals with no particular preference for risk bearing [32]. 

Two common, rational theories of decision making are used to explain individuals’ decision to purchase health insurance. From the perspective of the expected utility theory (EUT), in which consumers are uncertain whether or not they will become ill and unclear of the related financial consequences at the time of insurance choice [26], the purpose of purchasing insurance is, therefore, to reduce the uncertainty. The act of owning insurance increases certainty and allows the insured to reach a higher utility when care is needed. Based on the EUT, the demand for insurance reflects individuals’ risk aversion and demand for income certainty [33]. This theory is salient about the association between a household’s socio-economic status and insurance enrollment [34,35]. The other theory of decision making is prospect theory, which advocates that the choice of health insurance is about the prospects of gains and losses instead of level of uncertainty [26]. Individuals evaluate a prospect based on a simple gain–loss value from a reference point rather than the prospect’s effect on final wealth [36]. Individuals assume an optimal risk level for every expected gain or loss and weigh the probability of gains and losses to determine their choice [37]. Building off prospect theory, Hwang (2016) [36] studied low-to-moderate-income U.S. individuals and found that loss-averse individuals have a low ownership rate of long-term care insurance, supplemental disability insurance, and private health insurance. 

Although theoretical models predict that the demand for insurance should increase with individual risk aversion [38,39,40], empirical studies found the relationship to be inconclusive. Several studies suggested that higher risk propensity predicts stronger insurance demand [41,42] and a more risk-averse person may purchase less insurance for self-protection [43,44]. 

Risk propensity is an important determinant of decision making behavior in managers as well as customers [29]. Theoretically, risk aversion seems to be the main reason for the purchase of insurance. We assume that risk-averse individuals are likely to be aware of their risk attitude and its consequences; thus, they perceive protection of loss as necessary. 

**Hypothesis** **2.**
*Individuals with low risk propensity score (risk averse) are more likely to have LTCI or demonstrate future intention to own LTCI.*


### 2.4. Personality Traits

Risk propensity was found to be strongly rooted in personality [45]. Personality changes throughout the life span, but with more pronounced changes in young and old ages due to social demands and experiences [46]. Personality traits have been documented as sources of influence on the tendency to take or avoid risks, including achievement motivation [47], sensation seeking [48], and low self-control [49]. Cloninger (1987) suggested that individuals who are harm-avoiding are more inclined to plan ahead before they engage in an activity [50]. Personality plays a central role in consumer decision making, particularly in behavioral contexts related to health and well-being [51]. 

Among the Big Five personality traits, openness to experience showed reliable mean-level decreases with age [52,53,54]. Openness to experience has been linked to responsiveness to training. Individuals with higher openness take on a more open-minded and positive attitude during training and learn more as a result [55]. People high in this dimension are inclined to take chances. Thus, openness to experience can be seen as the cognitive counterpart to risk seeking, acceptance of experimentation, tolerance of uncertainty, change, and innovation [56]. Based on the characteristics of curiosity and novelty of openness to experience, we posit the following:

**Hypothesis** **3.**
*Individuals with higher openness to experience are less likely to purchase LTCI or demonstrate future intention to own LTCI.*


### 2.5. Life Experience 

Due to uncertainty about the unknown future health, insurance choice is not made based on utility alone but also with consideration of the individual’s expectation of factors such as their health status [57] or experience witnessing parental use of a nursing home [58]. Personal experience with illness and poor health has an effect on lifestyle decisions and increases the demand for health insurance [42]. These personal experiences could come from either an individual’s own perception of the degree of risk or from the opinion of those surrounding the individual [32]. Royal (2016) suggested that individuals more often purchase insurance following events or disasters to reduce their future risks [59]. Individuals weighted directly experienced outcomes more heavily than those experienced indirectly. Meyer (2012) also reported recent losses to be responsible for reductions in risk-taking behavior, particularly if losses were unprotected [60]. Earlier life experience with providing informal long-term care to others may alter one’s demand for LTCI, though these previous experiences may not be limited to one’s personal experience [29]. 

**Hypothesis** **4.**
*Individuals with personal experience of catastrophic diseases (or long-term caring experiences to sick family members or currently living with elders) are more likely to purchase LTCI or demonstrate future intention to own LTCI.*


## 3. Methods

### 3.1. Data and Sample 

This study is a cross-sectional and survey research. The inclusion criteria used included that participants must be at least 20 years old and have a minimum of two years of full-time working experience, in order to assure that the research samples were adults and basically financial independent. We surveyed 1474 nurses during December 2018 to February 2019. There were 1407 returned questionnaires, including 34 with incomplete data, with a 95.45% response rate. The final sample was 1373.

### 3.2. Measures 

#### 3.2.1. Dependent Variables

In this study, the dependent variable was a categorical variable with three levels: currently own private long-term care insurance (coded as 1, indicating an effective private LTCI), future intention to own LTCI (coded as 2, indicating future intention to buy LTCI and no previous purchasing experience), and others (coded as 3, indicating not currently own and no future intention to own). 

#### 3.2.2. Independent Variables 

##### Personal Discretionary Income (PDI)

We defined PDI as the amount of an individual’s income that is left for spending, investing, or saving after paying taxes and paying for personal necessities, such as food, shelter, and clothing. The variable was categorized as: ≤NTD (New Taiwan Dollar) 10,000; NTD 10,001~20,000; NTD 20,001~30,000; NTD 30,001~40,000; and ≥NTD 40,001. 

##### Risk Propensity

We applied the Risk Propensity Scale (RPS) developed by Meertens and Lion (2008) to measure the tendency to seek or avoid risks [61]. The scale contains seven items and is measured by a nine-point, Likert-type scale with 1 indicating totally disagree and 9 indicating totally agree. The alpha reliability (α) in this study was 0.70. Higher scores on the RPS indicated higher risk-seeking tendencies. 

##### Openness to Experience

We adopted the dimension of openness to experience from the Big Five Inventory Personality Test (BFI) [62] to assess participants’ traits. Openness to Experience contains 10 items, ranging from 1 (strongly disagree) to 5 (strongly agree). One sample item is “I see myself as someone who is original, comes up with new ideas.” The alpha reliability (α) was 0.74. This variable was coded as a binary variable, with 1 indicating high openness to experience (scores ranging from 40~50 indicated higher openness personality) and 0 for low openness to experience (scores ranged from 10~39). To ensure content validity, we performed translation and back translation for the risk propensity and openness to experience scales. 

##### Life Experience

We created three questions to represent life experience: (1) personal experience of catastrophic diseases: we asked participants whether he/she had previously encountered events of catastrophic diseases, with 1 indicating “yes” and 0 indicating “no”; (2) long-term caring experiences to sick family members: we asked participants whether he/she had experience providing long-term caring to sick family members, with 1 indicating “yes” and 0 indicating “no”; and (3) currently living with elders: “yes” coded as 1 and “no” coded as 0. 

#### 3.2.3. Control Variables

Additional factors identified by prior studies as key components in the decision to insure against LTC needs were included. These variables include age, number of children, marital status (1 = married, 0 = others), education level (high school, some college/vocational school/associate degree, bachelor’s degree, and graduate/professional degree), tenured employees (1 = yes, 0 = contract based), total tenure (years), and number of available caregivers (family, relatives, or friends). We inquired about participants’ willingness to accept caregiving responsibilities if provided with governmental cash allowance (1 = yes, 0 = no). Although women have been found to be more risk averse in health [63] and more likely to purchase LTCI [64], we did not control for gender as 97.9% of our sample were women. 

### 3.3. Statistical Analysis

To evaluate the relationship between personal discretionary income (PDI), risk propensity, personality, life experience, and LTCI ownership, we performed descriptive, bivariate, and multinomial logistic regression analyses. In descriptive analyses, we analyzed the mean (standard deviation) and frequency (percentage) for continuous and categorical variables, respectively. Bivariate analyses were used to examine the aforementioned four variables and control variables against the nominal dependent variable (currently own private LTCI, future intention to own LTCI, and no future intention to own). We then used multinomial logistic regression analyses to describe data and to explain the relationship between the nominal dependent variable and the independent variables (PDI, risk propensity, personality trait, and life experience). The nominal dependent variable included (1) currently own private LTCI, (2) future intention to own LTCI, and (3) no future intention to own LTCI. The odds ratios (OR) and associated *p*-values were reported. All analyses were performed with SPSS version 24.0 [65]. 

### 3.4. Ethical Consideration 

This study was approved by the Kaohsiung Veterans General Hospital (approval number: VGHKSI8-CT9-03), where the data were collected. All participants were provided with written, informed consent prior to the survey. 

## 4. Results

### 4.1. Sample Characteristics 

About 45% of our participants reported ownership of private long-term care insurance (LTCI). Respondents who expressed future intention to own were 56.2%, while respondents who can afford it but express no future intention to own LTCI were 16.2%. The average age was 37.78 years. Most respondents were bachelor graduates (76.4%), and half of the respondents were married (50.7%). Average number of children was 0.93, number of available caregivers (family, relatives, or friends) was 2.11. Approximately 6% of respondents experienced catastrophic diseases and 37.5% had previous experiences in providing long-term care to sick family members. The mean score of risk propensity was 3.26. The percentage of high openness to experience was 5.8%. Almost 94% respondents agreed that cash allowance would increase their willingness to take care of elders. Table 1 shows the descriptive analyses.

### 4.2. Bivariate Analyses

We also examined differences in participants’ characteristics for the three groups (currently own private LTCI, future intention to own LTCI, and no future intention to own). The currently own private LTCI group was older (38.66 years vs. 36.38 years) and had more children (1.03 vs. 0.82) in comparison to the no future intention to own group. The percentage of tenured employees was higher in the currently own private LTCI group (43.2%), followed by future intention to own LTCI group (39.4%). The percentage of married participants was higher in the currently own LTCI group (55.6%), followed by intention to own LTCI group (49.4%) and no intention to own (43.1%) groups, respectively. The currently own private LTCI group had the highest percentage in experience providing long-term care to sick family members (41.4%), followed by the future intention to own group (34.5%). The highest average score on the risk propensity scale was observed in the no future intention to own group (3.45). Table 2 presents the differences among the three levels of the dependent variable. 

### 4.3. Multivariate Analyses

Results from the multinomial logistic regression models showed that substantial monthly personal discretionary income (PDI) was the strongest predictor of ownership of LTCI. The odds of owning LTCI for those with PDI NTD 10,001~20,000, PDI NTD 20,001~30,000, PDI NTD 30,001~40,000, and PDI NTD ≥ 40,001 were 1.68 times, 2.09 times, 2.15 times, and 1.67 times higher than those with less than PDI NTD 10,000, respectively. The odds ratios were higher for individuals with PDI NTD 20,001~30,000, followed by those with NTD 30,001~40,000. Hence, Hypothesis 1 was supported. Additionally, married individuals were more likely to currently own LTCI (OR = 1.61, *p* = 0.04). 

Risk propensity was found to be a statistically significant predictor of currently own LTCI (0.81 times, *p* = 0.002) and future intention to own (0.79 times, *p* = 0.001). Individuals with high risk propensity scores are less likely to own or have intention to own private LTCI. Hypothesis 2 was, therefore, supported. Individuals who agreed on caregiving responsibilities when provided with cash allowances are 2.11 times more (*p* = 0.04) likely to have future intention to own LTCI than those who did not agree. Individuals currently living with elders (OR = 1.34, *p* = 0.08) or with experiences of catastrophic diseases (OR = 1.79, *p* = 0.09) showed weak significance in intention to own LTCI in the future; therefore, Hypothesis 4 was partially supported. Finally, openness to experience showed small effects in predicting both currently own private LTCI and future intention to own private LTCI (OR = 1.35, *p* = 0.08; OR = 1.34, *p* = 0.10, respectively). Hence, we concluded that Hypothesis 3 was marginally supported. Results from the multinomial logistic regression models are shown in Table 3.

## 5. Discussion 

Our study supports income as a crucial indicator to predict ownership of LTCI. Literature has documented that long-term care coverage from LTCI is not suitable for everyone, particularly for those with lower income [66]. Poor households are expected to become increasingly risk averse and, as they move closer to the poverty line, LTCI becomes less affordable [67]. On the other hand, individuals with higher personal discretionary income (PDI) demonstrated greater ability to purchase LTCI. Findings are consistent with previous research [27,28]. Particularly, PDI in the range of NTD 30,001~40,000 showed higher likelihood of LTCI ownership, indicating risk sharing is common among those with middle income. However, PDI was not found to be predictive of intention behavior. 

Lower risk propensity (risk averse) can predict individuals’ future intention to own LTCI. To reduce the level of exposed risk, risk-averse people show greater intention to purchase long-term care insurance. This is consistent with the concept of expected utility theory—the risk-taking individuals accept a certain level of uncertainty, while the risk-averse reduce uncertainty by buying insurance. Previous studies found different results regarding the association between high risk propensity and insurance demand [42,43]. Thus, further elaboration is needed to clarify the role of risk propensity in influencing insurance ownership. We did not find that openness to experience was predictive of ownership of LTCI, intention to own, or no intention to own, but it was for affordable insurance premiums. Future research may consider exploring other personality traits or examining this trait in a different study sample. 

Life experience may impact the choice of purchasing LTCI. Although previous studies have found a relationship between prior experience with long-term care and the decision to purchase long-term care [68], our study did not find personal experience of catastrophic diseases to be predictors of ownership of LTCI, future intention to own, or future no intention to own. In addition, long-term caring experience for sick family members was not found to be a predictor in our three models. This finding is quite different from Courbage and Roudaut’s study (2008), which found that having provided informal care to family positively affects the probability of purchasing long-term care insurance [69]. However, we did discover that participants who currently live with elders are more likely to hold an intention to own LTCI. Living with elders provides opportunity to observe elderly trajectories from different life stages—in turn, influencing individuals’ emotional responses and potentially increasing insurance demand [68]. Our study supports evidence of the linkage between living with elders and the demand for LTCI, as Ranyard and McHugh (2012) argued that emotional responses create a link between past risk and future insurance purchases [70]. 

Two control variables, age and tenured employees, significantly predict an individual’s future no intention to own when able to afford the premium. The effect size of age is marginal. Tenured employees are lifelong employed, implying that they are more likely to have better fringe benefits while working and welfare after retirement, hence, their tendency to purchase LTCI. Participants who agree to caregiving responsibilities if provided with governmental cash allowance are more likely to show intention to own LTCI. Allaire et al. (2016) also found people supporting any type of government initiative public or private LTCI were more likely to choose LTCI [27]. Supporting public LTCI (any type, such as cash allowance) does not reduce the likelihood of purchasing private LTCI. 

### Limitation

Although this study is an advance over previous studies of LTCI in its use of a comprehensive scope (including income, risk-taking propensity, personality trait, and life experience) to predict three different dependent variables (currently own private LTCI, future intention to own, able to afford but no future intention to own), several limitations remain. First, our sampled subjects were nurses, a population that earns an above-average monthly wage (US$ 2405 vs. US$ 1856 for nurses and the general population, respectively) [71] and frequently takes care of elderly patients, thus increasing awareness to risks of financial insecurity due to the future need of long-term care services. This may explain for the higher percentage of private LTCI ownership that was observed in the data. Moreover, our sample may be relatively homogenous in terms gender, income, education, and potentially personality and risk aversion traits by virtue of participants’ decision to become nurses. This may influence the generalizability of our findings. Second, as we were unable to account for consumer knowledge specifically related to LTCI, we could not distinguish the effect of consumer awareness on the purchase of LTCI. 

Additionally, this study did not account for premium changes and availabilities of policies to meet the demand of LTCI. As literature has demonstrated, these two factors from the supply side could impact the decision of purchasing LTCI [15]. We are aware that financial inducement may increase motivation to purchase LTCI. However, given that the government tax credits in long-term care were not effective until May of 2020, we believe that this is not a crucial factor that would drastically impact the present results. Future studies should, however, consider the factor of financial incentives when exploring similar contexts. Finally, LTCI plans provide broad and diverse coverages, though we did not specify in the survey questions what services in particular would be rendered. The service items may influence one’s choice to purchase LTCI. 

## 6. Conclusions

Individuals with higher PDI predict ownership of private LTCI. Higher risk-taking propensity is negatively related to future intention to own private LTCI. Participants who currently live with elders and who agree to caregiving responsibilities with government-provided cash allowance show intention to purchase LTCI. Compared to nontenured employees, tenured employees reported no future intention to own LTCI. Our results may provide policymakers with a better understanding of the forces driving demand in the private LTCI market as well as the accompanying implications for public LTCI. 

## Figures and Tables

**Table 1 healthcare-09-00102-t001:** Descriptive characteristics of participants.

	Mean (SD)	Frequency (%)
Gender		
Male		28 (2.0)
Female		1345 (98.0)
Education		
High school		10 (0.8)
College/associate’s degree		220 (16.0)
Bachelor		1050 (76.4)
Graduate/Professional		93 (6.8)
Age	37.78 (9.35)	
Total job tenure (years)	13.93 (9.62)	
Marital status		
Married		699 (50.7)
Unmarried		674 (49.3)
Numbers of children		0.93 (1.05)
Personal discretionary income (NTD)		
≤10,000		268 (19.7)
10,001–20,000		468 (34.4)
20,001–30,000		303 (22.3)
30,001–40,000		169 (12.4)
≥40,000		152 (11.2)
Tenured employee		
Yes		546 (39.9)
No		827 (60.1)
Currently living with elders		
Yes		740 (53.8)
No		633 (46.2)
Long-term caring experiences to sick family members		
Yes		520 (37.5)
No		853 (62.5)
Risk propensity scale	3.25 (1.14)	
Openness to experience	3.30 (0.47)	
High (40–50)	80 (5.8)	
Low (others)	1293 (94.2)	
Experiences of catastrophic disease		
Yes		87 (6.4)
No		1286 (93.6)
number of available caregivers (family, relatives, or friends)	2.11 (1.38)	
Agree to caregiving responsibilities if provided with cash allowance		
Yes		1289 (93.9)
No		84 (6.1)
Currently own private LTCI		
Yes		615 (44.8)
No		758 (55.2)
Future intention to own LTCI		
Yes		426 (56.2)
No		332 (43.8)
Affordable but future no intention to own LTCI		
Yes		123 (16.2)
No		635 (83.8)

**Table 2 healthcare-09-00102-t002:** Comparison of three groups in characteristics of participants.

	Currently Own LTCI (*n* = 615)		Future Intention to Own LTCI (*n* = 426)		Others (*n* = 332)		
	Mean (SD)	Frequency (%)	Mean (SD)	Frequency (%)	Mean (SD)	Frequency (%)	Chi-Square Tests/ANOVA
Gender							
Male		16 (2.6)		7 (1.6)		5 (1.5)	χ^2^(2, *n* = 1373) = 1.76, *p* = 0.415
Female		599 (97.4)		419 (98.4)		327 (98.5)	
Education							
High school		5 (0.8)		3 (0.7)		2 (1.6)	χ^2^(6, *n* = 1373) = 7.92, *p* = 0.244
College/associate’s degree		89 (14.5)		65 (15.3)		66 (15.4)	
Bachelor		481 (78.2)		322 (75.5)		247 (70.7)	
Graduate/Professional		40 (6.5)		36 (8.5)		17 (12.2)	
Age	38.66 (9.24)		37.27 (9.23)		36.38 (9.51)		F(2, 1370) = 7.00, *p* = 0.001 **. Post Hoc Tests 1 > 3 (Scheffe)
Total job tenure (years)	14.79 (9.59)		13.90 (9.63)		12.20 (9.47)		F(2, 1370) = 5.26, *p* = 0.005 ** Post Hoc Tests 1 > 3 (Scheffe)
Marital status							
Married		343 (55.6)		211 (49.4)		143 (43.1)	χ^2^(2, *n* = 1373) = 13.40, *p* = 0.001 **
Unmarried		272 (44.4)		215 (50.6)		189 (56.9)	
Numbers of children	1.03 (1.06)		0.87 (1.01)		0.82 (1.07)		F(2, 1370) = 5.08, *p* = 0.006 ** Post Hoc Tests 1 > 3 (Scheffe)
Personal discretionary income (NTD)							
≤10,000		92 (15.0)		93 (22.1)		83 (25.4)	χ^2^(8, *n* = 1360) = 25.11, *p* = 0.001 **
10,001–20,000		208 (33.9)		140 (33.3)		120 (36.7)	
20,001–30,000		155 (25.3)		85 (20.2)		63 (19.3)	
30,001–40,000		80 (13.1)		59 (14.0)		30 (9.2)	
≥ 40,001		78 (12.7)		43 (10.2)		31 (9.5)	
Tenured employee							
Yes		265 (43.2)		167 (39.4)		114 (34.3)	χ^2^(2, *n* = 1373) = 6.99, *p* = 0.030 *
No		350 (56.8)		259 (60.6)		218 (65.7)	
Currently living with elders							
Yes		326 (52.9)		242 (56.8)		160 (48.2)	χ^2^(2, *n* = 1373) = 2.38, *p* = 0.304
No		289 (47.1)		184 (43.2)		172 (51.8)	
Long-term caring experience for sick family members							
Yes		252 (41.4)		146 (34.5)		113 (34.0)	χ^2^(2, *n* = 1173) = 7.22, *p* = 0.027 *
No		363 (58.6)		280 (65.5)		219 (66.0)	
Risk propensity scale	3.20 (1.13)		3.19 (1.12)		3.45 (1.16)		F(2, 1370) = 6.16, *p* = 0.002 **. Post Hoc Tests 3 > 1, 3 > 2 (Scheffe)
Openness to experience							
High		41 (6.7)		26 (6.1)		13 (3.9)	χ^2^(2, *n* = 1373) = 3.06, *p* = 0.217
Low		574 (93.3)		400 (93.9)		319 (96.1)	
Experiences of catastrophic disease							
Yes		38 (6.2)		34 (8.0)		15 (4.5)	χ^2^(2, *n* = 1173) = 3.89, *p* = 0.143
No		577 (93.8)		392 (92.0)		317 (95.5)	
Numbers of relative/friend share responsibilities of supporting elders	2.2 (1.39)		2.05 (1.40)		2.04 (1.33)		F(2, 1370) = 2.33, *p* = 0.098
Agree to caregiving responsibilities if provided with cash allowance							
Yes		579 (94.1)		407 (95.5)		303 (91.3)	χ^2^(2, *n* = 1373) = 6.14, *p* = 0.047 *
No		36 (5.9)		19 (4.5)		29 (8.7)	

Note: * Indicates significance at the *p* < 0.05 level. ** Indicates significance at the *p* < 0.01. Note: *n* = 1373; LTCI = long-term care insurance.

**Table 3 healthcare-09-00102-t003:** Multinomial logistic regression result of the effects of indicators on LTCI ownership.

	Currently Own Private LTCI vs. Others		Future Intention to Own Private LTCI vs. Others	
	OR	*p* Value	OR	*p* Value
Age	1.02	0.14	1.01	0.68
Marital status	1.61	0.04 *	1.27	0.32
Education				
College vs. high school	0.80	0.80	0.66	0.66
Bachelor vs. high school	1.25	0.80	0.92	0.93
Graduate vs. high school	1.26	0.80	1.31	0.79
Numbers of children	0.93	0.54	0.95	0.71
Numbers of relative/friend share responsibilities of supporting elders	1.02	0.68	0.96	0.56
Tenured employee	1.01	0.98	1.12	0.57
Personal discretionary income (PDI)				
NTD 10,001~20,000 vs. ≤ 10,000	1.68	0.01 *	1.05	0.80
NTD 20,001~30,000 vs. ≤ 10,000	2.09	0.002 **	1.07	0.79
NTD 30,001~40,000 vs. ≤ 10,000	2.15	0.01 **	1.50	0.18
NTD ≥ 40,001 vs. ≤ 10,000	1.67	0.09 †	1.05	0.87
Risk propensity	0.81	0.002 **	0.79	0.001 **
Openness to experience	1.35	0.08 †	1.34	0.10†
Experiences of catastrophic diseases	1.12	0.74	1.79	0.09 †
Long-term caring experiences to elders	1.09	0.61	0.81	0.23
Current live together with elders	1.24	0.18	1.34	0.08†
Agree to caregiving responsibilities if provided with cash allowance	1.43	0.24	2.11	0.04 *
−2 Log likelihood	2474.154, χ^2^= 76.308 (*p* = 0.000 ***)			

Note: PDI: personal discretionary income (monthly, counted in New Taiwan Dollars). † Indicates significance at the *p* < 0.10 level. * Indicates significance at the *p* < 0.05 level. ** Indicates significance at the *p* < 0.01.

## Data Availability

The data presented in this study are available on request from the corresponding author. The data are not publicly available due to ethical guideline.

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
