# Peer review of "Private Long-Term Care Insurance Decision: The Role of Income, Risk Propensity, Personality, and Life Experience"

_healthcare, 2021, doi:10.3390/healthcare9010102_

Round 1

Reviewer 1 Report

The article provides an analysis useful for understanding the factors that motivate people to purchase Long-term care insurance. The results obtained can be useful for insurance companies. In general, the article is written in an understandable way, although there are several comments. Comments are made in the text of the attached copy of the manuscript.

Reviewer 2 Report

This paper uses a sample of nurses in Taiwan to examine individual correlates with the purchase and intention to purchase long-term care insurance. The key contribution of this study is its examination of additional individual characteristics, including personality type and an alternate measure of risk aversion not used in earlier studies. 

My biggest concern with this paper relates to the sample construction and outcome variables. Specifically, the paper uses separate models to examine the probability of 1.) Currently owning long-term care insurance 2.) Future intention to own long-term care insurance and 3) the ability to afford LTCi without any intention to buy. First, the inclusion criterion for each of these three models differs (or should differ), and the others are not clear about this in their methods. For instance, the first outcome can be estimates in the full sample. But the sample for the second outcome is presumably conditioned on not already owning LTCi. Based on the reported sample sizes in the Table 3, it appears the authors did condition in this manner, but this should be discussed explicitly in the methods. Furthermore, the authors might consider combining this outcome with actual LTCi ownership, either as a binary variable that can be examined with logistic regression (Pr(owning or intending to own LTCi), or as a categorical variable the can be modeled with a multinomial logistic regression.

The third outcome is most problematic for me. It seems odd to simultaneously model the ability to afford (based on a current income threshold) and the absence of an intention to buy insurance. I was initially concerned about a mechanical link between income as an independent variable and the dependent variable, but it appears the authors excluded income as an IV in the model for this outcome. This should have been specified in the methods section, but it is still not obvious to me how to interpret the estimates from this model. Notably, the estimates show the odds of not intending to own LTCI and being able to afford it vs. intending to own LTCi and not being able to afford it, not intending to own LTCi and being able to afford it, and not intending to own and not being able to afford it. The authors should drop this outcome and consider running their first to models on the subsample of respondents who meet the income threshold for affordability.

My other major concern with this paper relates to the lack of information necessary to contextualize LTCI purchase in Taiwan. In the US, this market is complicated by the existence of safety-net coverage in the form of Medicaid. Does Taiwan have a similar safety net that may change the income levels at which LTCI purchase makes economic sense and may crowd out demand for private LTCI? How are policies subsidized, if at all, in Taiwan? In the absence of greater information about the private LTCI market, I had a hard time interpreting results

Related, there was no discussion about how the affordability threshold discussed above was set. More information here is needed. my why this outcome  This should not be modeled on both the left and right hand sides of the equation. 

Finally, I think the authors need to do more to acknowledge that their study sample may severely limit generalizability. The authors make mention of this in the limitations, but the issue here goes beyond the study sample being more likely to own LTCI. Specifically, their sample is fairly homogenous in terms income, education, and potentially personality and risk aversion by virtue of their decision to become nurses. This lack of sample variation places this study at risk for being underpowered to detect effects of these traits that may in fact exist in the general population.
